# Learning Planning Abstractions from Language

**Weiyu Liu**[1*]**, Geng Chen**[1*]**, Joy Hsu**[1]**, Jiayuan Mao**[2†]**, Jiajun Wu**[1†]
[1] Stanford   [2] MIT

## Abstract

This paper presents a framework for learning state and action abstractions in sequential decision-making domains. Our framework, *planning abstraction from language (PARL)*, utilizes language-annotated demonstrations to automatically discover a symbolic and abstract action space and induce a latent state abstraction based on it. PARL consists of three stages: 1) recovering object-level and action concepts, 2) learning state abstractions, abstract action feasibility, and transition models, and 3) applying low-level policies for abstract actions. During inference, given the task description, PARL first makes abstract action plans using the latent transition and feasibility functions, then refines the high-level plan using low-level policies. PARL generalizes across scenarios involving novel object instances and environments, unseen concept compositions, and tasks that require longer planning horizons than settings it is trained on.

## 1 Introduction

State and action abstractions have been widely studied in classical planning, reinforcement learning, and operations research as a way to make learning and decision making more efficient (Giunchiglia & Walsh, 1992; Ghallab et al., 2004; Sutton et al., 1999; Li et al., 2006; Rogers et al., 1991). In particular, a state abstraction maps the agents' raw observation of the states (e.g., images or point clouds) into *abstract* state descriptions (e.g., symbolic or latent representations). Action abstraction constructs a new library of actions which can be later *refined* into raw actions (e.g., robot joint commands). "good" state and action representations are beneficial for both learning and planning, as state abstraction extracts relevant information about the actions to be executed, and action abstraction reduces the search space for meaningful actions that an agent could try.

Many prior works have explored methods for learning or discovering such state and action abstractions, for example, through bi-simulation (Givan et al., 2003), by learning latent transition models (Chiappa et al., 2017; Zhang et al., 2021), by inventing symbolic predicates about states (Pasula et al., 2007; Silver et al., 2021), or from language (Andreas & Klein, 2015; Corona et al., 2021). In this paper, we focus on the latter, and learn abstractions from language *for planning*. Fig. 1a shows the overall learning and planning paradigm of our system. Our goal is to construct a language-conditioned policy for solving complex tasks. Given a sufficient amount of demonstration data or online experience, a reinforcement learning agent could, in theory, learn a language-conditioned policy to solve all tasks. However, in scenarios where there is a significant amount of variations in the number of objects, initial object configurations, and planning steps, learning such policies is inefficient and at times infeasible. Therefore, we explore the idea of discovering a *symbolic* abstraction of the action space from language and inducing a latent state abstraction for abstract actions. This allows agents to *plan* at test time in an abstract state and action space, and reduces the horizon for policy learning by decomposing long trajectories into shorter pieces using the action abstraction.

In particular, our framework, *planning abstraction from language* (PARL) takes a learning and planning approach. First, illustrated in Fig. 1a, given paired demonstration trajectories and language descriptions of the trajectories, we recover a symbolic abstract action space, composed of *object-level* concepts and *action* concepts. Each object-level concept describes a property of an object (e.g., its category or color), and each action concept is a *verb schema* that takes objects as its argument (e.g., *clean*). These object and action concepts can be recombined in a compositional manner to form

---

*denotes equal contribution. † denotes equal advising. Project page: https://parl2024.github.io/.

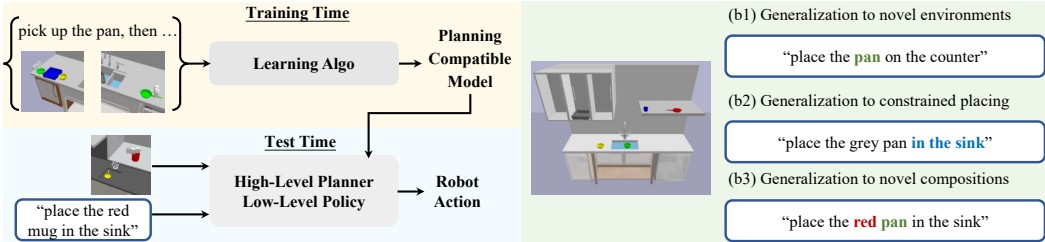

Figure 1: The overview of our training and testing paradigm, and different types of generalizations supported by our framework. (a) Given paired demonstration trajectories and language descriptions, our framework discovers an abstract action space and a latent state abstraction that supports planning for diverse language goals. (b) The example illustrates that our model can generalize to a new kitchen environment, generalize to a goal that requires reasoning about the geometries of the sink and grey pan, and generalize to the combination of concepts *red* and *pan* that is unseen during training.

the entire action space (e.g., *clean*(*blue bowl*)). Next, we learn four models based on the abstract action space: (1) a state abstraction function that maps raw observations to a latent space, (2) an abstract action transition model in the induced latent space, (3) an abstract action feasibility model that classifies whether the execution of an abstract action will be successful on a particular abstract state, and (4) a policy model that maps the current raw state and the abstract action to a raw action that the agent can directly execute in the environment. Unlike works that aim to simultaneously learn a symbolic action and state abstraction (Konidaris et al., 2018; Silver et al., 2021), our framework learns a *latent* state abstraction as it allows us to capture important geometric constraints in environments, such as whether the sink has enough space to place a particular object.

Such state and action abstraction enables us, *at test time*, to use *planning* in the abstract action space to solve tasks. In particular, during test time, given the high-level task description (e.g., clean the bowl), we first translate it into a corresponding abstract action to execute (i.e., *clean*(*bowl*)), and then search for an abstract action sequence that sets up the preconditions for the target action (e.g., by clean up other objects in the sink and turn on the faucet), and finally, execute the target action.

PARL enables various types of generalizations, and we highlight some of them in Fig. 1b. First, due to the adoption of an object-centric representation for both states and actions, PARL generalizes to scenarios with a different number of objects than those seen during training, and generalizes to unseen composition of action concepts and object concepts (e.g., generalizing from cleaning red plates and blue bowls to cleaning red bowls). Second, the planning-based execution enables generalization to unseen sequences of abstract actions, and even to tasks that require a longer planning horizon.

In summary, this paper makes the following contributions. First, we propose to *automatically* discover action abstractions from language-annotated demonstrations, without any manual definition of their names, parameters and preconditions. Second, our object-centric transition and feasibility model for abstract actions enables us to recover a latent state abstraction without the annotation of symbolic predicates, such as object relations or states. Finally, through the combination of our abstraction learning and planning algorithm, we enable generalization to different object numbers, different verb-noun compositions, and different planning steps.

## 2 RELATED WORK

Many prior works have explored learning abstract actions from imitation learning or reinforcement learning using the guidance of language. Language instructions can be given in the form of sequences of action terms (Corona et al., 2021; Andreas et al., 2017; Andreas & Klein, 2015; Jiang et al., 2019; Sharma et al., 2021; Luo et al., 2023), programs (Sun et al., 2020), and linear temporal logic (LTL) formulas (Bradley et al., 2021; Toro Icarte et al., 2018; Tellex et al., 2011). A common idea underlying these works is that language introduces a hierarchical abstraction over the action space that can improve the learning data efficiency and performance. However, they either focus on learning hierarchical policies or focus on pure instruction following tasks (Sharma et al., 2021; Corona et al., 2021). By contrast, our model leverages the action abstraction to *plan* (via tree search) in the abstract action space for better generalization to more complex problems.

Integrated model learning and planning is a promising strategy for building robots that can generalize to novel situations and goals. Specifically, Chiappa et al. (2017); Xu et al. (2019); Zhang et al. (2021); Schrittwieser et al. (2020); Mao et al. (2022) learn dynamics from raw pixels; Jetchev et al. (2013); Pasula et al. (2007); Konidaris et al. (2018); Chitnis et al. (2021); Bonet & Geffner (2020); Asai & Muise (2020); Silver et al. (2021); Zellers et al. assume access to the underlying factored states of objects, such as object colors and other physical properties. Our model falls into the first group where we learn a dynamics model from perception inputs (point clouds in our case) and the dynamics model operates in a latent space. The primary difference between our work and other method in the first group is that instead of learning a dynamics model at the lowest primitive action level, we learn a transition model at an abstract level, by leveraging language instructions.

Another line of work has focused on learning latent transition models and planning with them (Shah et al., 2022; Wang et al., 2022; Huang et al., 2023; Agia et al., 2023). These models focus on learning a latent transition model for a *given* set of primitive actions (by contrast, we discover these actions from language instructions), and their algorithm involves searching or sampling continuous parameters for their actions (e.g., finding grasping and placement poses). By contrast, our goal is to learn an abstract transition model in the latent space where the action set is purely discrete. This allows us to use simpler tree search algorithms for planning and furthermore, enables us to solve more geometrically challenging tasks such as object placements with learned feasibility.

A general goal of our system is to build agents that can understand language instructions and execute actions in interactive environments to achieve goals (Kaelbling, 1993; Tellex et al., 2011; Mei et al., 2016; Misra et al., 2017; Nair et al., 2022). See also recent surveys from Luketina et al. (2019) and Tellex et al. (2020). Furthermore, the abstract action representation can be viewed as options in hierarchical reinforcement learning (HRL; Sutton et al., 1999; Dietterich, 2000; Barto & Mahadevan, 2003; Mehta, 2011), and is related to domain control knowledge (de la Rosa & McIlraith, 2011), goal-centric policy primitives (Park et al., 2020), and macro learning (Newton et al., 2007). Our problem formulation is largely based on existing work on HRL, but we focus on discovering options from language and learning planning-compatible abstract transition models.

## 3 PROBLEM FORMULATION

We study the problem of learning a language-conditioned goal-reaching policy. In this work, we assume a fully observable environment, denoted as a transition model tuple $\langle \mathcal{S}, \mathcal{A}, \mathcal{T} \rangle$ where $\mathcal{S}$ is the state space*, $\mathcal{A}$ is the action space, and $\mathcal{T} : \mathcal{S} \times \mathcal{A} \to \mathcal{S}$ is the transition function. Each task $t$ in the environment is associated with a natural language instruction $L_t$. The instruction $L_t$ could either be a simple goal description (e.g., "clean the bowl") or a sequence of steps (e.g., "first pick up the red bowl from the countertop, then put it in the sink, and finally clean it"). The core insight of our approach lies in the assumption that each natural language description $L_t$ consists of abstract actions. Each abstract action $a' \in \mathcal{A}'$ can be factorized as $(w_1, w_2, \ldots, w_K)$, where $w_1$ is a verb symbol (e.g., "clean") and each remaining $w_i$ is an object argument to the verb, represented by a set of object-level symbols (e.g., "red bowl"). For example, the instruction "pick up the red bowl from the sink" can be factorized into $(pick\text{-}up, red, bowl, sink)$.

Our training data is composed of language-annotated demonstrations and interactions, each of which consists of state-action pairs and the corresponding $L_t$. We also assume that each trajectory has been segmented, such that the segments can be aligned to abstract actions in the instruction $L_t$. To allow the system to understand the feasibility of different actions, our training dataset also contains *failure* trajectories. In particular, the execution of some trajectories may lead to a failure, for example, in scenarios where the agent can not place the object in the target position, as the target space has been fully occupied by other objects. During test time, we evaluate our system on unseen instructions $L_t$ that contain a single goal action (e.g., "place the bowl into the sink"). Often the goal action is not immediately feasible (e.g., there are other objects blocking the placement); therefore the agent need to plan out additional actions before taking the goal action.

From the language-annotated data, our objective is to learn a policy with state and action abstractions. The abstractions essentially correspond to the abstract state space $\mathcal{S}'$, the abstract action space $\mathcal{A}'$, and the abstract transition function $\mathcal{T}'$. Our state abstraction function $\phi : \mathcal{S} \to \mathcal{S}'$ maps raw states to

---

*We assume that the raw state space is object-centric; that means, each raw state $s \in \mathcal{S}$ will be represented by an image or a point cloud, together with the instance segmentation of all objects in the scene.

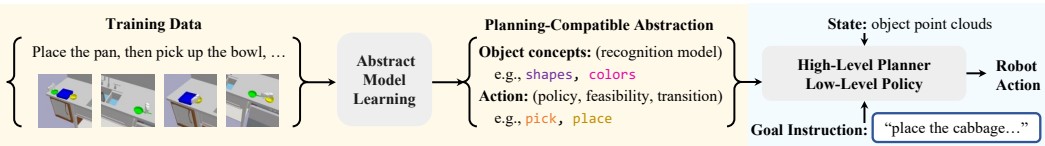

Figure 2: The overall framework of PARL. PARL takes paired natural language instructions and demonstration trajectories as inputs. It recovers object-level concepts such as shapes and colors, and action concepts from the natural language. It then learns a planning-compatible model for object and action concepts. At test time, given novel instructions, it performs a combined high-level planning and low-level policy unrolling to output the next action to take.

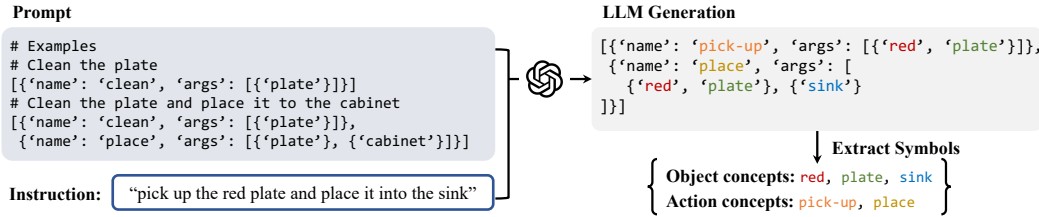

Figure 3: PARL prompts a pretrained large language model (LLM) to parse instructions into symbolic formulas. Next, we extract the object-level and action concepts from the formulas.

abstract states. Similarly, function $\mathcal{T}' : \mathcal{S}' \times \mathcal{A}' \to \mathcal{S}'$ learns an abstract transition model. To facilitate planning and execution with these abstract actions $a' \in \mathcal{A}'$, we additionally learn a feasibility model $f_{a'} : \mathcal{S}' \to \{0, 1\}$, which indicates whether an abstract action is feasible at the abstract state, and a low-level policy model $\pi_{a'} : \mathcal{S} \to \mathcal{A}$ that can refine each abstract action into primitive actions in $\mathcal{A}$ conditioned on the current environmental state.

## 4 PLANNING ABSTRACTION FROM LANGUAGE

Our proposed framework, *planning abstraction from language* (PARL), is illustrated in Fig. 2. PARL operates in three main stages: symbol discovery, planning-compatible model learning, and test-time planning and execution. We first *discover* a set of action and object concepts from the language instructions in the dataset by leveraging pretrained large language models (LLMs). Next, we *ground* these symbols using the demonstration data and interactions. In particular, we learn a planning-compatible model consisting of the state abstraction, transition and feasibility for abstract actions, and low-level policy. During inference, given state and action abstractions, we translate the instructions into symbolic formulas represented using the discovered action and object concepts, and use a search algorithm based on the learned transition and feasibility functions to generate plans and execute.

### 4.1 SYMBOL DISCOVERY

In the first stage, we consider all $L_t$ instructions in the training dataset and employ pretrained large language models (LLMs) to discover a set of symbols. Each action in $L_t$ is subsequently translated into a symbolic formula $(w_1, w_1, \ldots, w_K)$, which contain the discovered action and object concepts. Illustrated in Fig. 3, given the instruction $L_t$, we prompt LLMs to parse the instruction into a list of abstract actions, where each action consists of an action concept (e.g., *pick-up*), and a list of arguments, where each argument is represented as a set of object concepts (e.g., *red*, *plate*). We provide detailed prompts in Appendix A.1. In contrast to representing each action term as a textual sentence, the decomposed action and object concepts allow us to easily enumerate all possible actions an agent can take. This compositional approach can become especially useful when an agent needs to take additional abstract actions to achieve a language goal that is not immediately feasible.

### 4.2 PLANNING-COMPATIBLE MODEL LEARNING

In the second stage, we utilize the language-annotated demonstration data to *ground* the discovered action and object concepts on the raw environmental states. Specifically, we train a group of four models. First and foremost, we learn a state abstraction function $\phi : \mathcal{S} \to \mathcal{S}'$ that maps the raw state to the abstract state. Second, we learn an abstract transition function $\mathcal{T}'$ that models transitions in the abstract state space $\mathcal{S}'$. Third, for each abstract action $a'$, we learn a feasibility function $f_{a'}$

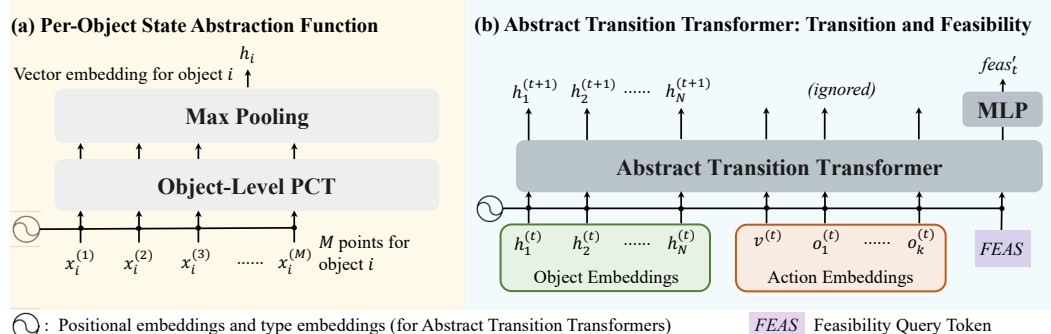

Figure 4: Neural network architectures for our planning-compatible models, composed of (a) an object-level PCT encoder for extracting state abstractions and (b) an abstract transition Transformer for abstract transition and the feasibility prediction.

that determines whether an action $a'$ is feasible in a given abstract state. Fourth, for each action $a'$, we learn a low-level policy function $\pi_{a'}$ that maps abstract actions to specific actions in the raw action space $\mathcal{A}$. In this section, we first describe an end-to-end model that unifies the state abstraction function, the abstract transition model, and the feasibility model. We present how the end-to-end model support plannning with abstraction in Section 4.3. We then present the details for the low-level policies in the Section 4.4. Below, we will describe our models for a raw state space $\mathcal{S}$ represented by 3D point clouds, but similar ideas can be easily extended to other representations such as 2D images, which we discuss in Appendix A.2. We present implementation details of our models in Appendix A.3.

**State abstraction function.** The state abstraction function $\phi$ is an object-centric encoder, which encodes the raw state from a given point cloud. Specifically, a state is represented by a list of segmented object point clouds $\{x_1, .., x_N\}$, where $N$ is the number of objects in the environment. Each object point cloud contains a set of spatial points and their RGB colors, i.e., $x_i \in \mathbb{R}^{6 \times M}$, where $M$ is the number of points in each point cloud. We encode each point cloud separately using a Point Cloud Transformer (PCT; Guo et al., 2021) to obtain latent features for the objects $\{h_1, \cdots, h_N\}$.

**Abstract transition model.** The abstract transition function $\mathcal{T}'$ takes the abstract state $s'_t = \{h_1, \cdots, h_N\}^{(t)}$ and the abstract action $a'_t$ as input, and predicts the next abstract state $s'_{t+1} = \{h_1, \cdots, h_N\}^{(t+1)}$. As illustrated in Fig. 4b, this function is implemented by an object-centric Transformer. The input to this function contains two parts: the abstract state representation for time step $t$ as a sequence of latent embeddings of the objects, and the encoding of the abstract action as a sequence of token embeddings. The function predicts the abstract state representation at time step $t + 1$, which is the sequence of output tokens from the Transformer that correspond to the objects. Recall that an abstract action is composed of a verb name $v$ (e.g., *place*), and a sequence of object-level concepts for each argument (e.g., *red plate* and *countertop*). We encode the factorized action $(w_1, w_2, \cdots, w_K)$ as a sequence of discrete tokens using learnable token embeddings. We use positional embeddings to differentiate each element of the input sequence to the Transformer encoder.

**Feasibility function.** One important purpose of the abstract transition model is to be able to predict the feasibility of a future action. For example, we want the agent to understand that it needs to pick up an object before moving the object; and it needs to ensure there is enough space in the target region such that it can place the object in the target region. We predict feasibility by integrating a binary feasibility prediction module $f_{a'}$ with the aforementioned object-centric Transformer. Illustrated in Fig. 4b, we append a feasibility query token to the combined sequence of object embeddings for $s'_t$ and action token embeddings for $a'_t$. The output of the Transformer encoder at this query position will be used to predict a binary label for the feasibility of executing $a'_t$ at $s'_t$. Specifically, we use another small Multi-Layer Perceptron (MLP) module on top of the Transformer for prediction.

**Training.** Our training data contains paired language instructions and robot interaction trajectories. For some of the demonstrations, the execution of the last action may cause a failure, therefore indicating an infeasible action. We use the successful and failed trajectories as positive and negative examples, respectively, to jointly train our state abstraction function, the abstract transition model, and the feasibility model.

In particular, each data point is in the form of the sequence $\{s_0, a'_0, \cdots, a'_{\ell-1}, s_\ell\}$, where $a'$'s are the abstract actions parsed by the LLM, $s_0$ is the initial state, and subsequent $s_{i+1}$'s are the raw environment state after the execution of each abstract action $a'_i$. For each data sequence, we first randomly subsample a suffix to obtain $\{s_k, a'_k, \cdots, a'_{\ell-1}, s_\ell\}$. Next, we use $\phi$ to encode the new initial state $s_k$ as $s'_k = \phi(s_k)$. Then we recurrently apply the Transformer-based abstract transition model $s'_{i+1} = \mathcal{T}'(s'_i, a'_i)$ and feasibility prediction model $\overline{feas}_i = f(s'_i, a'_i)$, for all $i = k, \cdots, \ell - 1$. For this data point, the loss value is defined as

$$\mathcal{L} = \sum_{i=k+1}^{\ell} \|s'_i - \phi(s_i)\|_2 + \sum_{i=k}^{\ell-1} \text{BCE}(\overline{feas}_i, feas_i),$$

where BCE is the binary cross-entropy loss, and $feas_i$ is the groundtruth feasibility label. $feas_i$ is true for all $i < \ell$, and is true for $i = \ell$ if the trajectory execution is successful (i.e., the last action is successful) or false otherwise. During training, we use stochastic gradient decent to optimize $\phi, \mathcal{T}'$, and $f_{a'}$, by randomly sampling a batch of data points.

### 4.3 Planning with Learned Abstractions

Based on the learned abstract transition model and feasibility function, we use a simple tree search algorithm to find abstract action sequences that can achieve the language goal. This is required when the given language goal only contains the last abstract action to achieve (e.g., *put the bowl into the sink*.) Specifically, the search algorithm acts in a breadth-first search (BFS) manner; it takes the current raw state $s_0$ and finds a sequence of high-level actions $(a'_1, \cdots, a'_K)$ such that $a'_K$ is the underlying abstract action in the language goal.

Given $s_0$, we first map it to a latent abstract state $s'_0$ using the state abstraction model. Then we enumerate all possible $a'_0 \in \mathcal{A}'$ (the first action) and compute their feasibility. Subsequently, the abstract transition model is applied to predict the subsequent abstract state $s'_1$ for all generated $a'_0$'s. By repeating this strategy recursively we can generate the final search tree. For each generated action sequence $a'_0, \cdots, a'_K$ and the corresponding feasibility scores $\overline{feas}_i$, we compute the feasibility of the entire sequence as $\min_i \overline{feas}_i$. The algorithm expands the search tree for a fixed horizon, and returns the sequence that ends with the specified last abstract action with highest feasibility score.

To accelerate search, the algorithm prunes the search space at every iteration by considering only a subset of abstract action sequences based on the feasibility scores of the sequence. This essentially resembles a beam-search strategy. As the computation for different states and actions can be parallelized, this algorithm can be efficiently implemented on a GPU. During execution, we use closed-loop execution, where we replan after executing each action using the low-level controller.

### 4.4 Low-Level Policy

Given the high-level plan computed by the search algorithm, our next step is to apply low-level policies $\pi_{a'} : \mathcal{S} \to \mathcal{A}$ that conditions on the current raw state $s$ to refine an abstract action $a'$ to a primitive action $a$ that the agent can directly execute in the environment. Our framework is agnostic to the specific implementation of the low-level policy. For experiments in the 2D gridworld with simplified navigation actions and object interactions, we adopt the FiLM method to fuse state and language information into a single embedding (Perez et al., 2018), then apply an actor-critic method to learn the policy. For experiments in 3D environments that require more precise end-effector controls for object manipulation, instead of outputting robot control commands, we ask the low-level policy to output either an object to pick, or an object to drop the current holding object on (essentially giving the robot a pick and place primitive). Both models are trained with data sampled from language-paired demonstrations and can predict actions directly from raw observations.

## 5 Experiments

We evaluate our models in two different domains: BabyAI (Chevalier-Boisvert et al., 2019), a 2D grid world environment, and Kitchen-Worlds (Yang et al., 2023), a 3D robotic manipulation benchmark.

### 5.1 BabyAI

**Environment.** The BabyAI platform features a 2D grid world, in which an agent is tasked to navigate and interact with objects in the environment given language instructions. Each simulated environment

| | Longer Steps | | | Novel Concept Combinations |
|---|---|---|---|---|
| | Key-Door | All-Objects | Two-Corridors | All-Objects |
| End-to-End BC | 0.00 | 0.00 | 0.00 | 0.00 |
| End-to-End A2C | 0.00 | 0.00 | 0.00 | 0.00 |
| Ours (high + low) | **0.45** | **0.39** | **0.52** | **0.27** |
| High-Level BC | 0.00 | 0.00 | 0.00 | 0.00 |
| High-Level A2C | 0.27 | 0.00 | 0.44 | 0.00 |
| Ours (high only) | **0.87** | **0.89** | **0.94** | **0.91** |

Table 1: Success rate of different models on BabyAI. We compare with behavior cloning and A2C baselines on two settings: (End-to-End) directly predicting low-level actions, and (High-Level) predicting high-level action tokens with oracle low-level policies.

contains four types of objects (key, door, ball, and box) with six possible colors. We focus on tasks involving two types of subgoals: 1) **Pickup**, which requires the agent to navigate to a target object specified by its color and type and then pick up the object; and 2) **Open**, which requires the agent to navigate to a colored door and open it with a key of matching color. The demonstration data are generated with the bot agent native to the platform. All models tested on this environment use a factorized observation space, which encodes the objects and their properties in each grid cell.

**Setup.** We evaluate models on three task settings: **Key-Door**, **All-Objects**, and **Two-Corridors**. In **Key-Door**, an agent needs to retrieve a key in a locked room. We design the environments such that the agent must first find a sequence of other keys to unlock intermediary doors before gaining access to the target key. Shown in Fig. 5a, for instance, to get a blue key behind a yellow-locked door, the agent first acquires a red key to unlock a red door, revealing a yellow key. The **All-Objects** setting is similar to Key-Door, except that the goal can be any object types (e.g., pick up the blue box, Fig. 5b).

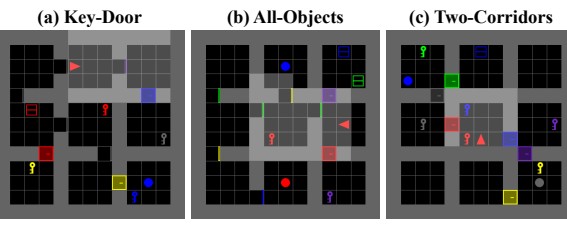

(a) Key-Door     (b) All-Objects     (c) Two-Corridors

Goal: pick blue key    Goal: pick green box    Goal: pick green key

Figure 5: We evaluate our models on diverse task settings created in the BabyAI environments.

Compared to Key-Door, All-Objects requires the models to accurately ground more visual concepts and efficiently plan with a larger action space. In the **Two-Corridors** setting, the agent needs to reach a target key that is at the end of a corridor (see Fig. 5c). To traverse each corridor, the agent needs to find keys to unlock intermediary doors. Each environment has two disconnected corridors, therefore requiring the agent to plan multiple steps into the future to find the correct path to the goal. We provide details for the settings in Appendix C.1.

Furthermore, we evaluate two types of generalizations on the aforementioned task settings. To test generalization to longer steps, we train the models on task instances that require at most 3 abstract actions to complete and test on task instances that can be solved with a minimum of 5 abstract actions. To test generalization to novel concept combinations, we withhold data involving *red key* during training and evaluate on goals involving this unseen concept combination. For all experiment, we evaluate on 100 task instances and report the average success rate.

**Baseline.** We compare our method with two learning-based baselines based on behavior cloning (BC) and advantage actor-critic (A2C). Both baselines use similar neural network backbones as our model, which processes the factorized observation with Feature-wise Linear Modulation (FiLM; Perez et al., 2018). The behavior cloning model is trained to predict the action to take from demonstrations with a cross-entropy loss, and the model is trained on the same demonstration dataset as our model. The A2C model has the exploration stage, where the agent can act itself in the environment and receive rewards. For a fair comparison with our bi-level planning and acting model, we studied two groups of models: End-to-End and High-Level. The End-to-End BC/A2C model is trained to directly predict low-level actions in the original action space (e.g., turn left, move forward, pick up an object), but compared to the baseline in BabyAI, we train it to solve tasks that require a significantly larger number of steps (e.g., the hardest benchmark in Chevalier-Boisvert et al. (2019) usually only involves unlocking one intermediate door before reaching the target). The High-Level BC/A2C model is trained to predict

| | Novel Environments | | Novel Concept Combinations |
|---|---|---|---|
| | All-Objects | Sink | All-Objects |
| Goal-Conditioned BC | 0.82 | 0.39 | 0.14 |
| Ours | **0.86** | **0.61** | **0.57** |

Table 2: Success rate of different models on Kitchen-Worlds. We compare Goal-Conditioned BC with our model on generalization to novel concept combinations and environments.

abstract actions and uses a manually scripted oracle controller to refine abstract actions to low-level commands. We provide implementation details for the baselines in Appendix B.1.

**Result.** We present the results on generalization experiments in Table 1. The End-to-End A2C failed to solve the tasks due to the large low-level action space and the long-horizon nature of our tasks. Since here we are only testing models on generalization to more complex tasks (they can all achieve nearly perfect performance on tasks seen during training), we found that self-explorations in the A2C model help the agent generalize to unseen scenarios. Our model (high+low) can efficiently explore the environment with learned abstraction. More importantly, we observe that our model (high+low) can generalize the learned dependencies between actions (e.g., to open a locked door, must first pick the corresponding key) to longer planning horizons, which is crucial for completing challenging **Two-Corridors** tasks. We also evaluate our model using the oracle low-level controller and observed a significant increase in performance after decoupling planning and low-level control. The same trend was not observed for the High-Level RL baseline, affirming the importance of learning a planning-compatible model instead of a policy. We also observe a very minimal performance drop when generalizing our models on new concept combinations. Since our model uses the network based on FiLM, it can decompose a language instruction to individual action and object concepts, therefore supporting compositional generalization to new combinations. We additionally study two variants of our method (high only) on generalization to longer steps for **All-Objects**. Without replanning, the success rate of the model decreased from 0.89 to 0.65. After replacing the FiLM network with a MLP to fuse information from language and observation, the success rate dropped to zero, suggesting that disentangling concepts are crucial for robust generalization. In Appendix D.1, we show examples of the generated plans for all three task settings.

## 5.2 KITCHEN-WORLDS

**Environment.** The Kitchen-Worlds (Yang et al., 2023) environments feature diverse 3D scenes and objects in a 3D kitchen environment. It requires the models to ground visual concepts on 3D observations and predict action feasibilities that are affected by geometric features of the objects. Each environment consists of objects from six categories (medicine bottle, water bottle, food, bowl, mug, and pan) and six colors (red, green, blue, yellow, grey, and brown). A robotic gripper is used to move objects around five types of storage units and surfaces (sink, left counter, right counter, top shelf, and cabinet). The simulation is supported by the PyBullet physics simulator (Coumans & Bai, 2017), which we also use to verify the feasibility of the grasps, object placements, and motion trajectories. We place six simulated cameras in the environment to capture the whole environment. RGB-D images and instance segmentation masks are rendered and converted to point clouds.

**Setup.** We test the models on two task settings: **All-Objects** and **Sink**. For both settings, we instantiate the environments with random scene layouts, furniture and object assets, and initial object placements. In **All-Objects**, the task involves placing the target object at a desired location. The main challenge of this task is to ground visual concepts involving diverse objects and locations, without explicit supervision (i.e., there is no classification label for object colors and categories). The optimal solutions involve one pick and one place action. The second **Sink** setting is designed to evaluate the models' abilities to reason about object shapes and environmental constraints. The solution could involve one to two pick-and-place actions. For example, shown in Fig. 7, given a sink that is occupied by a large object, the agent needs to first remove the object from the sink, then

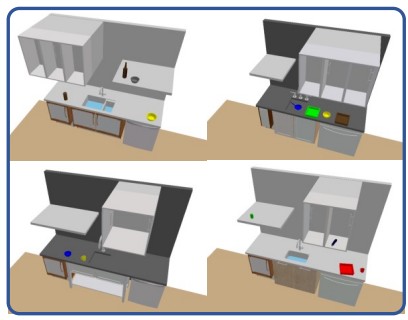

Figure 6: The kitchen environments with 3D furniture and objects.

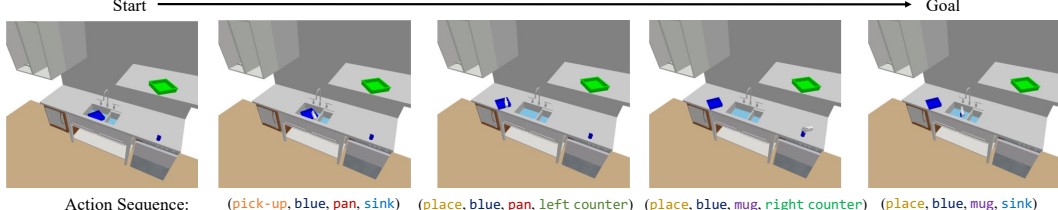

Figure 7: Given the language goal of placing blue mug in the sink, our model can reason about the geometry of the environment and perform planning with learned abstract latent space to predict the sequence of abstract actions. Each abstract action is expressed as a verb term and object arguments. For example, *(pick-up, blue, pan, sink)* corresponds to picking up the blue pan from the sink.

place the target object in the sink. Given enough space, the object can be directly placed in the sink. We provide details for the settings in Appendix C.2.

We conduct generalization experiments on the two task settings discussed above. We first evaluate whether the models can generalize to new environments with novel object instances. This is particularly important because the visual concepts are directly grounded on point clouds. We then evaluate whether the learned models can generalize to new task instances involving withheld object concepts, specifically *red bowl*. We evaluate 100 task instances and report the average success rate.

**Baseline.** We compare our method with Goal-Conditioned BC, which is trained to predict the sequence of abstract actions from a starting low-level state conditioned on a language goal. The model is implemented as a Transformer Encoder-Decoder, where the Encoder encodes object point clouds and the goal, and the decoder autoregressively decodes a sequence of abstract actions including the verb and object arguments; details are presented in Appendix B.2. We augment the training data by sampling start and end states, analogous to hindsight goal relabeling (Andrychowicz et al., 2017). During inference, the model predicts the next best action and replan after each step.

**Result.** Both Goal-Conditioned BC and our model perform well on generalizing to new object instances for **All-Objects**. Because the Goal-Conditioned BC model is trained on relabeled goals, it can fail to find the optimal plan for a given goal, therefore leading to long execution sequences or even time out. By contrast, our model, instead of imitating the demonstrations, learns the underlying abstract transition models. We observe a more significant performance difference between our model and the baseline on **Sink**. Our model can implicitly reason about the size and shape of the occupying objects in the sink to predict whether directly moving other objects into the sink is feasible. This type of reasoning is performed *implicitly in the latent space instead of requiring manually defined logical predicates*. This result demonstrates that our model is able to recover a state abstraction (e.g., whether an object can be placed in the sink) from paired language and demonstration data. In Appendix D.2, we show that our model is able to accurately predict feasibilities for abstract actions, even 5 steps into the future. Figure 7 illustrates one example where the feasible plan is found. Furthermore, we see strong performance from our models in generalization to novel concept combinations. These results demonstrate the advantage of our factorized state representation, where concepts of colors and object categories can be combined flexibly to support compositional generalization. We perform an ablation study by replacing our object-centric transformer with a FiLM network that operates on a scene-level abstract latent state and observed 6% decrease in success on the **Sink** setting. This result highlights the benefits of explicit object-centric representations for geometrically challenging tasks.

## 6 CONCLUSION

In this paper, we introduce *planning abstraction from language (PARL)*, a framework that leverages language-annotated demonstrations to automatically discover state and action abstraction. We learn transition and feasibility models for abstract actions that can recover a latent state abstraction without the annotation of symbolic predicates and generalize to unseen action-object compositions, and longer planning horizons. Currently, our method relies on segmented trajectories where the actions in language instructions are aligned with the low-level commands. Future work can leverage automatic segmentation of trajectories (Sharma et al., 2021; Luo et al., 2023). Another future direction is to integrate pretrained foundation models (Radford et al., 2021; Zhang et al., 2022; Wang et al., 2023) for object recognition with the factorized representation for generalization to unseen object concepts.

## ACKNOWLEDGMENTS

This work is in part supported by NSF RI #2211258, ONR MURI N00014-22-1-2740, ONR N00014-23-1-2355, ONR YIP N00014-24-1-2117, ASOSR YIP FA9550-23-1-0127, and Samsung.

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

# Supplementary Material for
# Learning Planning Abstractions from Language

The appendix is organized as the following. In Appendix A, we specify the prompts we use for PARL as well as PARL's model architecture in 2D and 3D. In Appendix B we discuss the baseline implementations, and in Appendix C we describe the environments and data collection strategy for BabyAI and Kitchen-Worlds. In Appendix D, we provide additional experiments. We discuss the limitations of our method and future directions in Appendix E.

## A  PARL

### A.1  PROMPTS FOR LLMS

We provide examples of the prompts we use for the GPT-4 LLM below. Recall that each natural language $L_t$ is parsed to one or multiple abstract actions. Each abstract action is represented as a sequence of symbols $(w_1, w_2, \ldots, w_K)$, where $w_1$ is a verb symbol (e.g., "clean") and each remaining $w_i$ is an object argument to the verb, represented by a set of object-level symbols (e.g., "red bowl").

```
Map each natural language instruction to symbolic
actions. Each symbolic action should contain the name and its arguments.

Example: Clean the plate.
Output: [{'name': 'clean', 'args': [{'plate'}]}]

Example: Clean the blue plate and place it into the sink
Output: [{'name': 'clean', 'args': [{'blue', 'plate'}]},
         {'name': 'place', 'args': [{'blue', 'plate'}, {'cabinet'}]}]

Example: Pick up the red plate and place it into the sink.
Output:  [{'name': 'pick_up', 'args': [{'red', 'plate'}]},
          {'name': 'place', 'args': [{'red', 'plate'}, {'sink'}]}]

Example: Pick up the blue mug from the sink, place it in the cabinet.
Then pick up the brown pan from the counter, and place it into the sink.
Output:  [{'name': 'pick_up', 'args': [{'blue', 'mug'}, {'sink'}]},
          {'name': 'place', 'args': [{'blue', 'mug'}, {'cabinet'}]},
          {'name': 'pick_up', 'args': [{'brown', 'pan'}, {'counter'}]},
          {'name': 'place', 'args': [{'brown', 'pan'}, {'sink'}]}]
```

Listing 1: Example GPT-4 prompt to map natural language instructions to symbolic formulas that our model consumes. Prompt context is in gray, input language descriptions are green, and generated word tokens are purple.

### A.2  MODEL ARCHITECTURE FOR BABYAI

We describe the model architecture for the BabyAI grid world below. The model comprises the state abstraction function, the abstract transition model, and the feasibility model. The model can be trained end-to-end with the objectives described in Section 4.2.

**State abstraction function.** The state abstraction function $\phi$ encodes the raw state $s_t$ to a latent feature map $s'_t \in \mathbb{R}^{H \times W \times C}$, where $H$ and $W$ are the height and width of the grid world, and $C$ is the dimension of the latent feature. In particular, the raw environment state $s_t$ is represented by a $13 \times 13$ grid, where each cell stores three integer values representing the indices of the contained object, its color, and its state (e.g., locked). A convolutional neural network (CNN) is used to process the raw environment state and map it to the latent feature map.

**Abstract transition model.** The abstract transition function $\mathcal{T}'$ takes the abstract state $s'_t \in \mathbb{R}^{H \times W \times C}$ and the abstract action $a'_t$ as input, and predicts the next abstract state $s'_{t+1} \in \mathbb{R}^{H \times W \times C}$. For a given abstract action, we encode its factorized representation $(w_1, w_2, \cdots, w_K)$ as a sequence

of discrete tokens using learnable token embeddings. To predict the next latent state $s'_{t+1}$, we use a total of $K$ FiLM layers (Perez et al., 2018) to modulate the latent feature map representing the current abstract state $s'_t$. Each FiLM layer is conditioned on the latent token of the symbol $w_i$ in the factorized action.

**Feasibility function.** The feasibility function predicts whether an abstract action $a'_t$ can be performed in an abstract state $s'_t$. Similar to the abstract transition model, $K$ FiLM layers are used to fuse the latent tokens of the abstract action with the latent state. The resulting feature map is then flattened and passed to a Multi-Layer Perceptron (MLP) module with two linear layers to predict the binary feasibility for the abstract action.

## A.3 Model Architecture for Kitchen-Worlds

Below we provide details for the end-to-end model based on an object-centric Transformer that unifies state abstraction, abstract transition, and feasibility prediction.

**State abstraction.** For a given raw environment state $s_t$, The state abstraction function encodes the corresponding sequence of $N$ segmented object point clouds $\{x_1, \cdots, x_N\}^{(t)}$, where each point cloud contains $M$ number of $xyz$ points and their RGB colors, i.e., $x_i \in \mathbb{R}^{6 \times M}$. We use a learned encoder $h_x$ to encode each object point cloud separately as $h_x(x_i)$. This encoder is built on the Point Cloud Transformer (PCT; Guo et al., 2021). We treat the encoded object latents as the latent abstract state, i.e., $\{h_x(x_1), \cdots, h_x(x_N)\}^{(t)} = \{h_1, \cdots, h_N\}^{(t)}$

**Tokenization.** The abstract transition model and feasibility function are both conditioned on the given abstract action $a'_t$. We convert the factorized representation of an abstract action $(w_1, w_2, \cdots, w_K)^{(t)}$ to a sequence of latent tokens using learnable word embeddings. Specifically, we convert $w_i$ to latent token $h_w(w_i)$, for $i = 1, \cdots, K$.

**Auxiliary embeddings.** We use a learned embedding for the query token $h_q$. We use a learned position embedding $h_{pos}(l)$ to indicate the position index $l$ of the object point clouds, action tokens, and query token in input sequences to the subsequent transformer. We use learned type embeddings $h_{type}(\tau)$ to differentiate the object embeddings ($\tau = 0$), action embeddings ($\tau = 1$), and query embedding ($\tau = 2$).

**Object-centric transformer.** The object-centric transformer serves two purposes. First, given a latent abstract state $s'_t = \{h_1, \cdots, h_N\}^{(t)}$ and the abstract action $a'_t$ as input, and the transformer predicts whether the abstract action is feasible at this time step. Second, given the latent abstract state and a feasible abstract action, the transformer predicts the next abstract state $s'_{t+1} = \{h_1, \cdots, h_N\}^{(t+1)}$.

To predict feasibility, the input to the transformer consists of the object part $e$, the action part $c$, and query part $q$. Specifically, these are

$$
\begin{aligned}
e_i &= [h_i; h_{pos}(i); h_{type}(0)], \\
c_i &= [h_w(w_i); h_{pos}(i - N); h_{type}(1)], \\
q &= [h_q; h_{pos}(i - N - K); h_{type}(2)],
\end{aligned}
$$

where $[; ]$ is the concatenation at the feature dimension. The transformer takes as input the concatenated sequence $\{e_1, \cdots, e_N, c_1, \cdots, c_K, q\}^{(t)}$. The output token of the transformer at the query position is fed into a small Multi-Layer Perceptron (MLP) to predict the binary feasiblity label for the action $a'_t$.

To predict the next abstract state, the transformer ignores the query part of the input using the attention mask, and only takes in $\{e_1, \cdots, e_N, c_1, \cdots, c_k\}^{(t)}$. The output tokens of the transformer at the position of the objects are taken to be the latent abstract state for the next time step $\{h_1, \cdots, h_N\}^{(t+1)}$.

**Batching.** To batch plan sequences with different lengths and with different number of objects, we use zero embeddings at the padding positions and attention masks for the transformer to avoid the effect of these padded inputs. We do not compute loss for padded actions in plan sequences.

**Parameters.** We provide network and training parameters in Table 3.

| Parameter | Value |
|---|---|
| Number of points for object point cloud $M$ | 512 |
| Max plan lengths | 6 |
| Max number of objects $N_{max}$ | 11 |
| Length of factorized action $K$ | 4 |
| PCT point cloud encoder $h_x$ out dim | 352 |
| Word embedding vocab size | 23 |
| Word embedding $h_w$ | learned embedding |
| Word embedding $h_w$ dim | 352 |
| Position embedding $h_{pos}$ | learned embedding |
| Position embedding $h_{pos}$ dim | 16 |
| Type embedding $h_{type}$ | learned embedding |
| Type embedding $h_{type}$ dim | 16 |
| Transformer number of layers | 12 |
| Transformer number of heads | 12 |
| Transformer hidden dim | 384 |
| Transformer dropout | 0.1 |
| Transformer activation | ReLU |
| Loss | Focal ($\gamma = 2$) |
| Epochs | 2000 |
| Optimizer | Adam |
| Learning rate | 1e-4 |
| Gradient clip value | 1.0 |
| Batch size | 32 |
| Learing rate warmup | Linear ($T_{end} = 20$) |
| Learning rate decay | Cosine annealing ($T_{max} = 2000$) |

Table 3: Model Parameters

## B    BASELINES

### B.1    RL MODELS FOR BABYAI

We compare our method with two RL baselines based on Advantage Actor-Critic (A2C; Mnih et al., 2016). The **End-to-End RL** baseline takes in a partial and egocentric view of the environment represented in a $7 \times 7\times$ feature map and directly predicts one of the seven low-level actions (i.e., turn left, turn right, move forward, pick up an object, drop an object, toggle, terminate). The model uses a convolutional neural network to process the raw environment state and use FiLM layers (Perez et al., 2018) to condition the model on the input language instructions. The **High-Level RL** baseline predicts high-level abstract actions instead of low-level actions. The predicted high-level abstract actions are executed using an oracle controller provided by the BabyAI simulator. This model observes the raw environment state as a $13 \times 13 \times 3$ feature map and predicts an integer index representing the selected abstract action. For both baselines, a non-zero reward is credited to the agent only when the agent reaches the language goal. The reward is adjusted based on the number of low-level steps the agent takes to encourage efficient goal-reaching behavior.

### B.2    GOAL-CONDITIONED BEHAVIOR CLONING FOR KITCHEN-WORLDS

This baseline takes in the current raw environment state $s_t = \{x_1, \cdots, x_N\}^{(t)}$ and the goal abstract action $a'_{goal} = (w_1, w_2, \cdots, w_K)$ as input, and predicts the next abstract action to execute $a'_t = (w_1, w_2, \cdots, w_K)^{(t)}$. The model design of this baseline is related to transformer-based models (Chen et al., 2021; Liu et al., 2022; Mees et al., 2022) but different in the specific implementation we choose for our domain. This baseline uses the same object point cloud encoding, action tokenization, position embeddings, and type embeddings as our model, which are discussed in Appendix A.3. Different from the Transformer encoder that is used by our model, this baseline uses the Transformer encoder-decoder architecture. The encoder encodes the concatenated sequence of the object latents

for $s_t$ and action latents for $a'_{goal}$, and the decoder autoregressively decodes each element $w_i$ of the abstract action $a'_t$.

# C  ENVIRONMENTS

## C.1  BABYAI

Below we present details for the data collection procedures and evaluation setups of the task settings **Key-Door**, **All-Objects** and **Two-Corridors** below. The involved concepts are listed in Table 4.

| Type | Value |
|------|-------|
| Verb | Pick-up, Open |
| Object | Key, Door, Ball, Box |
| Color | Red, Green, Blue, Yellow, Grey, Purple |

Table 4: BabyAI Concepts

**Key-Door.** To collect training data for this task setting, we vary the environments by randomizing the placements of the doors for the 9 rooms in the grid world. During initialization, we recursively lock each room and place the key of the locked door in another room. As such, to solve the task, the agent must first find a sequence of other keys to unlock intermediary doors before gaining access to the target key. For each scene, we put in 3 keys for 3 locked doors and 3 distractor objects. For each scene, we collect a demonstration trajectory by first searching for the sequence of high-level actions to reach the goal using a task planner and then mapping the high-level actions to low-level actions, which can be executed by an oracle controller. In total, we collect 100,000 demonstration trajectories. For evaluation, the same randomized procedure is used to initialize the testing scenes. A trial is considered successful only when the agent reaches the language goal within a limited budget of 200 low-level steps.

**All-Objects.** The environments are randomized using the same procedure described above. Different from **Key-Door**, this task setting requires the agent to locate a target object that can be any of the following object types: key, ball, and box. Each scene contains 3 keys for 3 locked doors, 3 distractor objects, and 1 target object. In total, we collect 100,000 demonstrations using the combination of the high-level task planner and the low-level oracle controller. For evaluation, the same randomized procedure is used to initialize the testing scenes. A trial is considered successful only when the agent reaches the language goal within a limited budget of 200 low-level steps.

**Two-Corridors.** To collect training data for this task setting, we vary the environments by randomizing the placements of the doors for the 9 rooms in the grid world. We initialize each environment by constructing corridors such that the agent can traverse each corridor by unlocking intermediary doors and that the two corridors are disconnected from each other. Each corridor consists of 3 locked doors with 3 matching keys. Each scene contains a total of 6 key objects and 3 distractor objects. We collect 100,000 demonstrations using the combination of the high-level task planner and the low-level oracle controller. For evaluation, the same randomized procedure is used to initialize the testing scenes. A trial is considered successful only when the agent reaches the language goal within a limited budget of 200 low-level steps.

## C.2  KITCHEN-WORLDS

Below we present details for the data collection procedures and evaluation setups of the task settings **All-Objects** and **Sink** below. For both settings, the involved concepts are listed in Table 5.

**All-Objects.** To collect training data for this task setting, we randomly initialize the environments with different 3D models and placements of the furniture (e.g., cabinet, counter, and sink). Examples of the randomized environments are shown in Figure 6. For each scene, we randomly place up to 4 objects in the scene with randomized initial positions. For each scene, we collect three trajectories each containing 6 abstract actions by randomly choosing a feasible action to execute at each step and simulating with the Pybullet simulator. To collect failure cases, at each step, we also randomly

| Type | Value |
|------|-------|
| Verb | Pick, Place |
| Object | Bowl, Mug, Pan, Medicine Bottle, Water Bottle, Food |
| Location | Left Counter, Right Counter, Top Shelf, Sink, Cabinet |
| Color | Red, Green, Blue, Yellow, Grey, Brown |

Table 5: Kitchen-Worlds Concepts

execute an action and record whether the action is successful or failed. This procedure gives us failed trajectories with different lengths. In total, we collected interaction trajectories in 3920 scenes. For evaluation, the same randomized procedure is used to initialize the testing scenes. If the model fails to complete the goal abstract action in 5 steps, we consider the trial to be unsuccessful. Note that for this task setting, the optimal solution involves two abstract actions — a pick and a place action.

**Sink.** To collect training data for this task setting, we randomly initialize the environments as described above. For each scene, we randomly place up to 4 objects in the scene and choose a random subset of the the objects to place in the sink. For each scene, we collect three trajectories each containing 6 abstract actions. At each time step, we choose a sink-related action with 50% chance (e.g., pick an object from the sink or place an object into the sink). In total, we collected interaction trajectories in 3360 scenes. For evaluation, if the model fails to complete the goal abstract action in 5 steps, we consider the trial to be unsuccessful. For this task setting, the optimal solution can involve up to four abstractions.

# D ADDITIONAL RESULTS

## D.1 BABYAI QUALITATIVE EXAMPLES

We provide generated plans in Figure 8. Our model is able to accurately generate multi-step plans for diverse environments and language goals.

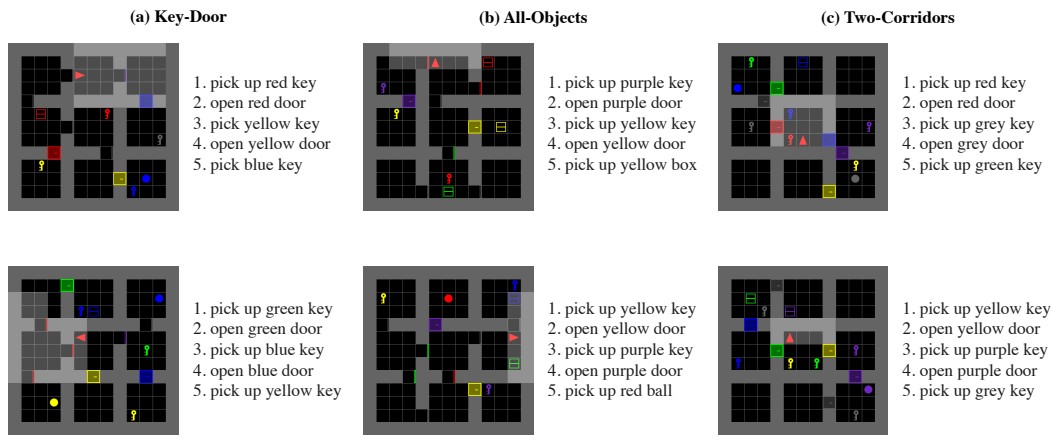

Figure 8: Plan predictions for different task settings. Each figure visualizes the initial state of the environment and the location of the agent (represented by the red arrow). On the right of each figure, the sequence of planned abstract actions is listed. The last abstract action in each list is the goal abstract action in the given language instruction.

## D.2 FEASIBILITY PREDICTION

We present a breakdown of feasibility prediction performance below. In general, we observe that our model is able to accurately predict feasibilities for abstract actions, even 5 steps into the future. The model performs well for both the task setting that requires grounding diverse visual concepts (i.e., **All-Objects**) and the one that requires geometric reasoning (i.e., **Sink**). We observe a decrease

in performance when the model generalizes to a new concept combination that is never seen during training. However, even in this challenging case, the average F1 score is still above 90% on the testing data.

| Task Setting | Generalization | Train | Test | | | | | | |
|---|---|---|---|---|---|---|---|---|---|
| | | Avg. | Avg. | $k$ | $k+1$ | $k+2$ | $k+3$ | $k+4$ | $k+5$ |
| All-Objects | New Env. | 99.70 | 98.17 | 99.00 | 98.51 | 97.79 | 97.32 | 96.45 | 98.89 |
| Sink | New Env. | 99.84 | 98.01 | 98.13 | 98.55 | 98.10 | 97.38 | 97.63 | 97.01 |
| All-Objects | New Concept Comb. | 91.97 | 90.08 | 88.74 | 90.36 | 87.67 | 90.24 | 88.68 | 99.01 |

Table 6: Feasibility prediction F1 scores for Kitchen-Worlds environments. For each setting, the scores on the training data and testing data are presented. For the testing data, we further present the feasibility prediction performance for abstract actions at different future time step $a_i$ given only current environment state $s_k$, for $i = k, \cdots, k+5$.

## E    LIMITATIONS

For the 3D environment, we assume the instance segmentation is provided so that we can extract object point clouds. Future work can explore the use of unsupervised object discovery (Sajjadi et al., 2022; Wang et al., 2021).

We assume that the input language instructions can be parsed into symbolic formulas following a fixed form. We further assume that the input language instructions are complete. Future work should relax these assumptions and operate on more diverse forms of language instructions.

In this work, we learn state abstraction and action abstract from language-annotated demonstrations. In order to scale to a large set of action and object concepts, our planning-compatiable models can be integrated with pretrained vision-language representations (Ma et al., 2023; Karamcheti et al., 2023) to bootstrap the learning of visual groundings and geometric reasoning.

