# OpenReview forum: "Learning Planning Abstractions from Language"
_ICLR.cc/2024/Conference — ICLR 2024 poster_

### Official Review · Reviewer_NUXY · 2023-11-01

**Soundness:** 3 good
**Presentation:** 3 good
**Contribution:** 2 fair
**Rating:** 6
**Confidence:** 4

**Summary:**

This work proposes a framework that learns state and action abstractions for planning. It does this by leveraging demonstrations with corresponding language annotations. These demonstrations are used to discover actions, which in turn is used to generate state abstractions. Finally, low-level policies are also learned corresponding to the high-level actions. The evaluation is done on two domains and the results show generalization wrt environments and objects.

**Strengths:**

1. The paper is well written. Except for the algorithm description, other parts like motivation, problem formulation, etc., are explained nicely.
2. The approach seems to be novel.

**Weaknesses:**

1. Other related approaches:
* There are related approaches that use language to guide the abstraction process. E.g., Peng et al., the difference is that human input. Here, this paper gets it in the form of language-annotated task descriptions.
* Approaches like LIV (with PointCLIP instead of CLIP) can learn a latent representation, which can be used for planning.

Peng et al., Learning with Language-Guided State Abstractions.

LIV: Ma et al., LIV: Language-Image Representations and Rewards for Robotic Control.

CLIP: Radford et al., Learning Transferable Visual Models From Natural Language Supervision.

PointCLIP:  Zhang et al., PointCLIP: Point Cloud Understanding by CLIP.


2. Reproducibility:
* I am not sure how reproducible the work is. There are a large number of details that are swept under the rug. And without an algorithm, it gets difficult to follow the paper. The supplementary material is also not submitted.
* The inputs are not clear.

3. Experimental Evaluation:
* I would suggest performing experiments for the accuracy of the feasibility function.
* The experiments from the grid-like BabyAI setup are not convincing of generalization. The paper claims to withhold "red key" in training, but they can learn the model agnostic to such properties. So, this is more of a verification that their approach works. But as we can see, the accuracy for novel concept combinations is only 91\%.

Minor points:
* Incorrect citation: I do not believe Silver et al. learn (invent) new predicates as stated in the last two lines of page 1.

**Questions:**

1. Who provides the examples for prompting in Fig. 3 left?
2. The training is performed on how many tasks? Was the environment structure the same for all of them? Or was it changed in between tasks? If it was changed, was it ensured that the test environment configuration was not present in the training set?
3. What is the reason for not achieving 100\% accuracy for novel concept combinations?

---

> ### Author Response · Authors · 2023-11-21
> **Response Part 1**
>
> **R4.1:** Related approaches for deriving state abstraction.
>
> Thank you for the suggestion! We agree that these methods can potentially help with state abstraction. However, we want to emphasize the key differences between these related approaches and our method, and therefore highlight our novelties.
>
> The vision-language representation LIV is trained on the objective such that (1) neighbouring frames in a video are close by in the embedding space, (2) their distances to the language goal smoothly decrease. These properties are useful for language-conditioned manipulation; however, we note that the representation is only trained on and evaluated on short-horizon interactions (e.g., open microwave, put hat on the bottle, and pineapple in the black pot). These interactions are analogous to a single abstract action in our definition. In comparison, our approach learns a state and action abstraction that is amenable to generating plans that involve multiple high-level actions.
>
> The key idea of Peng et al. is to identify task-relevant features of a visual scene conditioned on the input language. The abstraction process is done by extracting a list of objects along with their properties (e.g., silver pan, blue pot, and red square) and then using an LLM to determine which objects are relevant. Compared to this method, we do not assume that groundings of the object concepts are known, and show that our model can learn to reason about these concepts from language-annotated demonstrations. More importantly, the key idea of our approach is that the state abstraction should be latent. We illustrate that our model is able to predict that placing an object into the sink is not feasible if the sink is full. We do not explicitly symbolize "full" because it depends on the size of the object and the available space in the sink. Discretizing a raw state into explicit symbols, a process named textualization by Peng et al., prevents the method from performing fine-grained geometric reasoning.
>
> Finally, we want to highlight the challenges of applying CLIP and PointCLIP to our domain. CLIP mainly aligns images with textual descriptions of the visual contents. This is different from our abstract state representation which is tightly coupled with actions. In addition, existing research has found that CLIP behaves as a bag of words. This suggests that CLIP and its extension PointCLIP would not be able to infer the sink is currently full such that "placing a pan into the sink" is not feasible.
>
> A promising approach is to finetune these vision-language representations. We are currently running an experiment where we replace the point-cloud-based encoder with the LIV image encoder. Because our original data is collected with point clouds, we are currently recollecting the multi-view RGB image data to train the model. We will provide an update soon.
>
> **R4.2:** Details for the model.
>
> **A:** Thanks for the suggestion. We have included a new section (Appendix A) in the supplementary to document details of our models and training details. In addition, we have included new sections in the supplementary to document details of the baselines (Appendix B) and environment settings (Appendix C). We have also included more visualizations of the environment and the generated plans by our algorithm in Appendix D. We will also release the code.
>
> Regarding your particular question on inputs:
> - In BabyAI, the input to all methods is a 2D grid encoded with one-hot per-grid features (colors, object categories, agent's facing directions, etc.). This follows the original BabyAI paper [D1].
> - In Kitchen-World, the input to all methods is segmented 3D point clouds. In particular, the scene is composed of a set of objects; there is a colored point cloud for each object.
>
> [D1] Chevalier-Boisvert et al. BabyAI: A Platform to Study the Sample Efficiency of Grounded Language Learning. ICLR 2019.
>
> **R4.3:** Accuracy for feasibility function.
>
> **A:** Thank you for the great suggestion! We include a new evaluation on the accuracy of the feasibility function in the Kitchen-World environment in Appendix D (which is also copied down below). We break down the evaluation across different generalization environments and different prediction horizons. In short, our model is capable of generating highly accurate feasibility predictions.
>
> | Task Setting      | Generalization       | Train  | Avg.   | $k$    | $k+1$  | $k+2$  | $k+3$  | $k+4$  | $k+5$  |
> | ----------------- | -------------------- | ------ | ------ | ------ | ------ | ------ | ------ | ------ | ------ |
> | All-Objects       | New Env.             | 99.70  | 98.17  | 99.00  | 98.51  | 97.79  | 97.32  | 96.45  | 98.89  |
> | Sink              | New Env.             | 99.84  | 98.01  | 98.13  | 98.55  | 98.10  | 97.38  | 97.63  | 97.01  |
> | All-Objects       | New Concept Comb.    | 91.97  | 90.08  | 88.74  | 90.36  | 87.67  | 90.24  | 88.68  | 99.01  |

---

> > ### Author Response · Authors · 2023-11-21
> > **Response Part 2**
> >
> > **R4.4:** Validity of concept generalization.
> >
> > **A:** Thank you for bringing up this point. We would like to clarify that the grounding of concepts (e.g., red and key) is learned from data instead of hard-coded. Therefore, the grounding for both action objects (i.e., the "policies" and the feasibility models) and the object concepts (i.e., the "recognition" models) could have errors. Therefore, these results should be interpreted as: our model generalizes better than baselines that simply learn a mapping from states and texts to actions, rather than that our model would generalize perfectly.
> >
> > **R4.5:** LLM prompting details.
> >
> > The examples in the prompt are provided by humans. We use 2 examples to prompt the LLM. We included our prompts in the updated Appendix A.1.
> >
> > **R4.6:** Variation of environments for Kitchen Worlds.
> >
> > **A:** For Kitchen-Worlds, the training is performed on around 10,000 training trajectories (please see Appendix C.2 for specific numbers for different task settings). As shown in Figure 6, the environment is varied across tasks by changing the assets for the furniture and the placement of the furniture. Each environment is generated with a random seed; we made sure that the random seeds used to generate the testing environments are different from the random seeds used for training. We have included these details in Appendix C.2.

---

> > > ### Author Response · Authors · 2023-11-22
> > > **Update on experiment with pre-trained vision-language representation LIV**
> > >
> > > We are happy to update you that we have implemented a new goal-conditioned behavior cloning baseline based on the pre-trained LIV representation (*GC-BC-LIV*). The baseline is trained on the demonstration data for the *Sink* setting and evaluated on 100 testing scenes (the training and testing splits are the same as other methods). *GC-BC-LIV* achieves a 9.6% success rate; in comparison, the goal-conditioned behavior cloning baseline based on point clouds (*GC-BC-PC*) achieves a 39% success rate, and our model achieves a 61% success rate. We also notice that *GC-BC-LIV* converges slower than *GC-BC-PC* in training. We hypothesize that there are two key challenges in directly applying LIV to our tasks: 1) LIV is useful for short-horizon manipulation tasks but provides less meaningful signals useful for predicting future high-level action outcomes, which is critical for long-horizon manipulation tasks, and 2) *Sink* tasks require fine-grained geometric reasoning that is hard to capture in the representation pre-trained on 2D images (e.g., if there is enough space in the sink to place a target object into). In summary, this experiment highlights that our state abstraction is unique and novel because it is planning-compatible and aware of 3D geometry.
> > >
> > > We provide details for *GC-BC-LIV* below. To train the model, we recollected the demonstration data with multi-view images. Recall that we placed 6 simulated cameras in the environment to capture RGB-D images for different parts of the kitchen scene. Instead of converting these RGB-D images to object point clouds, we now use the pre-trained LIV image encoder to convert the RGB images to image embeddings. We use the pre-trained LIV text encoder to convert the language goal (e.g., "place the yellow mug in the sink") to a text embedding. Then, we use a similar transformer encoder-decoder as described in B.2 to encode the input image embeddings and text embedding, and decode the next abstract action.

---

> > ### Comment · Reviewer_NUXY · 2023-11-22
> >
> > Thank you for the response and updates to the paper. I think I have better clarity on the differences in the proposed approach as compared to the alternative approaches I suggested.
> >
> > 1. I would suggest adding citations for "existing research has found that CLIP behaves as a bag of words."
> >
> > 2. Why do you think the accuracy increases for $k+5$ as compared to $k+1$ for all-objects in the new table you provided? Is it significant?

---

> > > ### Author Response · Authors · 2023-11-22
> > > **Response to follow-up questions**
> > >
> > > Thank you for your response and suggestion!
> > >
> > > 1. Recent work by Yuksekgonul et al. [D2] provides evidence for our argument that CLIP behaves as a bag of words.
> > >
> > > 2. Regarding the relatively high accuracy at $k+5 $ for the **All-Objects** task setting when generalizing to **New Concept Combinations**, we believe this result does not suggest a consistent pattern. Two key reasons may cause the performance difference in predicting feasibility at different future steps. First, predicting feasibility for early steps (e.g., $k$) requires the model to recognize which objects appear in the environment and their locations (e.g., to predict whether “pick red bowl from left counter” is feasible requires the model to localize the red bowl) while predicting feasibility for later steps (e.g., $k+5$) can leverage dependencies between the abstract actions (e.g., “place red bowl in the cabinet” is likely to be possible if the previous action is “pick red bowl from the cabinet”). Second, we want to highlight that this particular result comes from the **New Concept Combinations** generalization. In this case, we believe the model happens to perform well on the withheld data at this epoch. We use early stopping based on validation loss for all models. We have checked the results to confirm that the performance converges to a lower accuracy when the model continues training, which is expected.
> > >
> > > [D2] Yuksekgonul et al. When and Why Vision-Language Models Behave like Bags-Of-Words, and What to Do About It?. ICLR, 2023

---

> > > > ### Comment · Reviewer_NUXY · 2023-11-23
> > > >
> > > > Thanks a lot for prompt response. I will increase the score to 6 to reflect the updated change after author discussion.

---

> > > > > ### Author Response · Authors · 2023-11-23
> > > > >
> > > > > Thank you for increasing the score. We appreciate your suggestions and comments that have helped us improve the paper.

---

### Official Review · Reviewer_V9Cw · 2023-11-01

**Soundness:** 3 good
**Presentation:** 3 good
**Contribution:** 3 good
**Rating:** 8
**Confidence:** 3

**Summary:**

The paper introduces a framework for learning state and action abstractions from language-annotated demonstrations. The abstract actions and states are use to train a transition model in the latent space to learn the feasilbility of newer latent actions. This allows agents to generalize actions learned from language to longer, unseen tasks.

**Strengths:**

**Originalty:** The paper investigates a problem is not tackled in the literature but can realistically exist. The paper is a novel and creative framework for addressing this problem.

**Clarity:** The paper is well-writtten.

**Significane:**  This work has the potential to be impactful in language-based agent interactions. Furthermore, the framework can be adapted to other sequential planning domains.

**Quality:** The problem described is well-motivated. The approach to addressing the problem is laid out clearly and simply and it reads reasonably. The model framework is creative and intuitive. The experimental design is sound and makes sense to test their claims and results support the claims made by the authors.

**Weaknesses:**

I don't have any major gripes. However, I found the description of the experimental domains lacking. Particularly I am not totally clear on the difference between the key-door and two-corridor environments.

**Questions:**

No questions.

---

> ### Author Response · Authors · 2023-11-21
> **Response**
>
> **R3.1:** Difference between key-door and two-corridor.
>
> **A:** Thank you for bringing this up. We have added a detailed discussion of the different task settings in Appendix C. In short, two-corridor is one specific kind of key-door environment and two-corridor is designed to be a harder task than randomly sampled key-door environments on average.
>
> In particular, in key-door, the connectivities between nearby rooms are randomly sampled. Therefore, it is likely that the required number of high-level actions is small or there is only one possible high-level action that is feasible at a state. In two-corridor, we specifically designed an environment where there are two corridors so that the planning length is long (maximum 7 high-level actions if the goal is to reach the room at the very end), and that at each given state, there will be at least two feasible actions. Therefore, in order to achieve the goal within a limited number of steps, the agent must unroll its learned transition model to select which corridor to enter.

---

> > ### Comment · Reviewer_V9Cw · 2023-11-22
> >
> > I acknowledge that I have read the author's comments. My score remains the same.

---

### Official Review · Reviewer_B6My · 2023-11-01

**Soundness:** 2 fair
**Presentation:** 3 good
**Contribution:** 2 fair
**Rating:** 5
**Confidence:** 3

**Summary:**

The paper proposes a framework for solving problems in sequential-decision making by combining LLM-generated high-level abstract actions, imitation learning and a low-level policy by a framework-agnostic traditional RL agent.

The pipeline in more detail is that given a prompt in human language which defines “a language goal”, an LLM decomposes to a verb and corresponding nouns and adjectives (e.g. ‘place’, ‘bowl’, ‘green’), with the assumption that these prompts can always be decomposed to this format. After this,

- a state abstraction function is learned that can identify the objects in the environment

- an abstract transition model is learned which predicts the next state given the current abstract state and high-level action. This model also has a feasibility component that predicts whether a future action can accomplish the language goal

- A breadth-first search algorithm selects the shortest sequence of actions that accomplishes the language goal

- Finally, a low-level policy is applied according to the sequence of high-level actions. These policies are learned with traditional RL

The paper tests the method on two environments: BabyAI (three task setting) and Kitchen-Worlds (two task settings), compare against low-level (regular) and high-level (when the agent has access to the defined ) RL in the former and Goal-Conditioned BC in the latter.

**Strengths:**

Originality: the paper proposes a novel way to solve sequential decision making problems by combining LLM prompting, imitation learning and traditional reinforcement learning.

Quality: the paper places the work in the literature very well, comparing the differences between previous works and mentions future work. The problem formulation is mostly clearly written up.

Clarity: The paper is mostly well-written and apart from a few inconsistencies, easy to understand.

Significance: its originality could be considered significant.

**Weaknesses:**

I have three main reservations:

- There is no available code, no experiment details (chosen hyperparameters, tuning) about the algorithm or the baselines and as such the results are not reproducible

- The experiments themselves, the results, and the metrics are described in a very high level without details which does not allow the reader to indeed verify how well they support the claims.

- Scalability: as the number of actions, objects and their combinations increase, the necessary training data size increases intractably (combinatorial explosion). I am concerned that this approach might be feasible for simple problems only due to its inherent limitations.

Furthermore, there are a few things that are unclear to me which could be further weaknesses. (I’ve listed the questions in the next section.)

My initial rating is due to the above reasons. I would be willing to increase the score if the above concerns are addressed adequately.

Clarity issues:
The notion of “tasks” is not defined in the problem formulation. I understand this is not easy to do, but including it would make the paper stronger. The expression “language goal” is also used a few times throughout the paper, without definition (and I believe tasks = language goals)

4.2 third paragraph third sentence. Did you mean to say something along the lines of “maps the abstract state representation at the current step and an abstract action $a’$ to the next abstract state $s’_{t+1}$” ?

Typos/style issues that did not affect the score:

Introduction
First paragraph; A “good” state and action representations -> Remove “A”
“As the state abstraction extracts relevant information about the actions to be executed” no need for the first “the”, and could you back this up with some sort of example in brackets, ideally with a citation (I can sort of guess what you mean, but I find this a bit too vague and unclear)

Third paragraph: “an particular object” -> a particular object

4th paragraph: “that setups” -> that sets up

4.2

Second paragraph:
“Which encodes the raw state from a given point cloud”: from -> to
“By applying a pooling operation for the point cloud” for -> to

Third paragraph:

“The abstract transition function $\tau’$ takes the following form. It maps [...] -> remove “takes the following form. It”

5th paragraph

“We appends” -> we append

“Of the Transformer encoder at this query possible will be used” -> no need for possible?

4.3
1st paragraph: last sentence is not needed, or its information content should be moved towards the end of the second paragraph. E.g. “we can generate the final tree, the leaves of which are $a’_K$ which correspond to the underlying abstract action in the language goal”.

2nd paragraph: subsequence -> subsequent

searchs ->searches

5
EXPERIMENT -> EXPERIMENTS

5.2

2nd paragraph: “designed to evaluate models’ abilities” -> missing the after evaluate

Citation for the Goal-Conditioned BC baseline is missing.

6

“A framework that leverage” -> leverages

**Questions:**

In the state abstraction function, how do you know when to stop training to have the right number of point clouds? Or else, do you assume that the number of objects in the environment is known beforehand? Is this end-to-end trained? (I think the paper would benefit if these points were made clearer there.)

What kind of pooling operation do you use? Max or average? And what is the dimensionality of the per-object latent feature?

How is the generalization success rate actually calculated? What does it mean that an agent “fails to solve the tasks”? I am assuming given the allowed number of steps?

For the loss of the feasibility prediction model, do you use the L_2 norm?

This is just notation, but in the reinforcement learning literature s’ usually denotes the next state, and $\hat{s}$ is used for an approximate state. Why did you decide to change this convention?

---

> ### Author Response · Authors · 2023-11-21
> **Response**
>
> **R2.1:** Implementation and experiment details.
>
> **A:** Thank you for the suggestion. We have included new sections in the supplementary to document details of our models (Appendix A), baselines (Appendix B), and environment settings (Appendix C). We have also included more visualizations of the environment and the generated plans by our algorithm in Appendix D. We will also release the code.
>
> **R2.2:** Scalability as number of concepts increases.
>
> **A:** Thank you for bringing up this point. We agree that scalability would be an issue for both learning and planning when the number of actions and objects increases. However, we should point out that this is _exactly_ the reason we learn a **compositional** space of candidate actions. In particular, we use LLMs to decompose instructions into steps, and steps into verbs and their object arguments. Therefore, our method can generalize to unseen compositions of concepts (see Table 1 and Table 2, the "novel concept composition" columns).
>
> Planning with a compositional space of possible actions would be another challenge, for all methods. In our paper, we have used a simple tree search algorithm with action feasibility pruning to generate plans. For more complex scenarios, more sophisticated algorithms, such as using symbolic planning tools [B1] or learning high-level policies to guide the search as in AlphaGo [B2] would be possible. We have tried a similar version with learned high-level policies; however, in practice, we find this policy barely generalizes to longer plans and hurts the performance.
>
> [B1] Helmert, Malte. "The fast downward planning system." JAIR (2006): 191-246.
> [B2] Silver, David, et al. "Mastering the game of Go with deep neural networks and tree search." nature 529.7587 (2016): 484-489.
>
> **R2.3:** Definition of task and language goal.
>
> **A:** Thanks for the suggestion. We have updated our problem formulation section to include definitions of tasks. Your understanding is correct that in this paper, each task $t$ in the environment is associated with a natural language instruction (the language goal) $L_t$.
>
> **R2.4:** Details for state abstraction function.
>
> **A:** Thank you for pointing these out. We have made these points more clear in the revision. Here are the direct answers to your questions:
> - **Point clouds**: As stated in Sec 4.2 State abstraction function, we assume access to segmented object point clouds. The number of objects in each scene is $N$ and is known.
> - **End-to-End training**: The state abstraction function $\phi$, the abstract transition model $\mathcal{T}'$, and the feasibility prediction model $f_{a'}$ are trained together end-to-end.
> - **Pooling operations**: We use max pooling and the per-object latent feature has a dimension of 352. We provide in Appendix A a list of hyperparameters for the model.
>
> **R2.5:** Details for metric.
>
> **A:** Thank you for bringing this up. We have added evaluation details in Appendix C. In summary, for BabyAI, models need to reach a goal within 200 low-level steps; otherwise, it's considered as a failure. For Kitchen-Worlds, models need to reach a goal within 5 steps; otherwise, it's considered as a failure. For all experiments, we evaluate on 100 task instances and report the average success rate.
>
> **R2.6:** Abstract transition model clarification.
>
> **A:** Your understanding is correct! We meant to say that the abstract transition model takes in the current abstract state $s'\_t$ and abstract action $a'$, and predicts the next abstract state $s'\_{t+1}$. We have updated the text to clarify this point.
>
> **R2.7:** Notation clarification.
>
> **A:** Thank you for the suggestion. We have chosen to use $s_t$ and $s'_t$ to represent the raw state and the abstract state at step $t$, respectively. Since we always use the subscript $t$ there are no ambiguities. However, we do agree that this will potentially create confusion for readers. We plan to use different symbols $x$ and $s$ to represent raw states and abstract states in the final version.
>
> **R2.8:** Loss clarification.
>
> **A:** Yes, we use L_2 norm. We have updated the equation to highlight this.

---

> > ### Comment · Reviewer_B6My · 2023-11-23
> >
> > Dear Authors,
> >
> > I will also need more time to revisit the revised version of the paper and the details in the response. Thank you for your response and the updates to the paper.

---

### Official Review · Reviewer_Q2SJ · 2023-11-02

**Soundness:** 2 fair
**Presentation:** 2 fair
**Contribution:** 2 fair
**Rating:** 3
**Confidence:** 3

**Summary:**

This paper presents an RL agent that solves the problem by utilizing symbolic abstraction and object-centric representation learning. Given a problem and a goal description for the environment, the proposed method uses LLM as a parser to translate the goal description (or instruction) as a collection of action predicates (verb and objects combination), where those action predicates are used as action abstraction (options or skills in hierarchical reinforcement learning).

Training requires human demonstration annotated with an action predicate. Given the demonstration data, the proposed method trains several functions. State abstraction function takes in the segmented output of a point cloud transformer to disentangle pixel input and it provides a latent space state representation. Action transition model takes in the latent abstract state and symbolic encoding of action predicates. Feasibility function predicts whether an abstract action is applicable in the current state.

After training necessary functions with annotated demonstration data, the planning stage uses the state abstraction function to get a latent state and utilize the feasibility function to select applicable actions. Planning with symbolic action predicates is done by brute-force search over all actions in the problem. Last,  given a high-level plan, the low-level policy is trained.

**Strengths:**

* Originality: The novel aspect of the presented method is combining symbolic planning with action predicates extracted from natural language goal descriptions or instructions and latent space representation learning with point cloud transformers. The abstract planning is done at the symbolic level, and the abstract state transitions are tracked with object-centric representation learned from segmented pixel data.
* Quality: The overall description of the method is easy to understand and the experiment was conducted on two types of the environments.
* Clarity: Figures help understanding the overall approach
* Significance: I think this is an interesting work that integrating many things to work.

**Weaknesses:**

* Originality: Individual components are existing approaches and the originality is on bring those components and implement an agent to solve mini-grid and kitchen world problems.
* Quality: Due to missing details, it is difficult to assess the quality.
* Clarity: There are many missing details in the paper.
* Significance: The comparison is made only against a simpler baselines (end to end RL and behavior cloning).

**Questions:**

### General questions
1. The title is “learning planning abstractions from language.” In the paper, the role of the LLM is parsing an instruction sentence to extract action predicates and objects. The remaining part of the work is independent of language models or language. The parsing could have been done manually or other methods. I cannot see the rationale of using LLM, other than demonstrating that LLM can do the parsing. What is “learned” from language?

2. What if the goal description or instruction did not reveal enough information to extract required high-level actions? Then, should we collect demonstrations following the derived high-level actions?

3. The instructions in the paper are quite simple sentences to parse. What is the longest abstract plan needed to solve the problem? How many abstract actions were needed?

### Section 4
4. In section 4.2, how the model was trained given annotated demonstrations? Is it learned per each abstract action? How many trajectories were given to the training process? How did you train Point Cloud Transformer for mini-grid environment and kitchen world environment? Can you present the details on the training of models?

5. In section 4.3, planning is done with BFS search. If the feasibility prediction fails, how did you handle the error? Does a set of actions derived from LLM parser always guarantee to solve a problem? How can you ensure the action space can solve all problems in the test set? Or a human demonstrator should create trajectories that solves problem given the action predicates?

6. In section 4.4, low-level policy is trained using an actor-critic algorithm. Can you present the details?

### BabyAI experiments.

7. The report on BabyAI experiments shows that baseline will not completely fail in all problems. The presented paper and the report also used similar low-level policy algorithm and neural network architectures. It also offers imitation learning experiments. Can you make some comparison with those baselines? What are the high-level actions extracted from LLM and what are the plans found for the problems? Can you present the sample efficiency or training/test performance?

[1] Hui, D. Y. T., Chevalier-Boisvert, M., Bahdanau, D., & Bengio, Y. (2020). BabyAI 1.1. arXiv preprint arXiv:2007.12770.

### Kitchen Worlds experiments.
8. How many high-level actions were annotated and trained to solve this environment?  From the description in the experiment section, the length of the plan is mostly one or two. Could you present details on the high-level plan and the low-level policies?

---

> ### Author Response · Authors · 2023-11-21
> **Response Part 1**
>
> **R1.1:** Parsing language with LLMs.
>
> **A:** Thanks for the suggestion. Following your suggestion, we would like to clarify how we are "learning from language" and "how we use language models."
>
> Our goal in the paper is to recover from language a symbolic abstract action space, composed of *object-level* concepts (such as colors and shapes) and *action* concepts (e.g., pick up). These object and action concepts can be recombined in a compositional manner to form the entire action space. In particular, we learn the grounding of these concepts: we learn recognition models for objects and policies for actions from paired demonstrations and texts, then use them in a planning algorithm to achieve a goal specified in language. In short, our method learns object and action concepts from language, and uses these concepts in language for planning.
>
> In order to decompose natural language instructions into individual concepts, we leverage large language models to parse sentences. We agree with the reviewer that the parsing can be done by other methods as well; we choose to use LLM to perform parsing due to its simplicity and accuracy. We have added example prompts we use for GPT-4 in Appendix A.1.
>
> **R1.2:** Incomplete language instructions.
>
> **A:** Thank you for bringing up this point. In our paper, the LLM is only used to generate the high-level goal (e.g., reaching a target object); the generation of high-level action sequences is performed by our planning algorithm, which takes environmental states as its input (for example, if reaching the target requires opening particular doors, the planner will handle such situations.)
>
> For scenarios where a user did not fully specify their goals, additional clarifications would be needed. This could be implemented by, for example, generating questions to the user. We did not consider such scenarios, therefore, we added a discussion of this limitation in Appendix E.
>
> **R1.3:** Details for abstract plans and actions.
>
> **A:** Thanks for bringing this up. We briefly discuss the abstract plans in the Setup subsections for BabyAI (Sec 5.1) and Kitchen-Worlds (Sec 5.2) due to the page limit. We have now added more details in Appendix C. We summarize the details below.
>
> For the BabyAI experiment, the most challenging case requires 5 abstract actions to solve. This is a nontrivial task because the models are only trained on task instances that require at most 3 abstract actions to complete. In addition, there are strong dependencies between the actions (e.g., the agent must first find the key to unlock an intermediary door before gaining access to the next key). Our empirical results show that baselines obtain significantly worse performance than our model.
>
> For the Kitchen-Worlds experiment, the most challenging case requires 4 abstract actions to solve (please see the Setup subsection in Sec 5.2). Similar to the BabyAI experiment, the abstract actions also have strong dependencies (e.g., the agent needs to first pick the object from the sink, place the object in empty space, then pick up the target object, and finally place the target object in the sink). In comparison to CALVIN [A1], which is a widely-used language-conditioned benchmark for long-horizon manipulation in 3D environments, our task is more challenging because (1) we significantly vary the environment layouts and (2) at test time, our goal instruction only contains a single goal action, and the agent needs to plan out additional subgoals before the target action; in contrast, CALVIN provides step-by-step instructions for solving tasks with 1-5 steps.
>
> [A1] Mees et al. Calvin: A benchmark for language-conditioned policy learning for long-horizon robot manipulation tasks. RAL, 2022.

---

> ### Author Response · Authors · 2023-11-21
> **Response Part 2**
>
> **R1.4:** Details for training (Section 4.2).
>
> **A:** We present training details in Sec 4.2. We have updated the text to further clarify the training procedure. Here we summarize the main points:
>
> - **Data**: Our model is trained on sequences of paired states and abstract actions, where each is in the form of $s\_0, a'\_0, s\_1, \cdots, a'\_{\ell-1}, s\_\ell$. At each iteration, we first randomly sample a suffix from a demonstration trajectory to obtain $s\_k, a'\_k, s\_{k+1}, \cdots, a'\_{\ell-1}, s\_\ell$. Next, we use $\phi$ to encode $s\_{k}$ as $s'\_{k} = \phi(s\_{k})$, and recurrently apply the Transformer-based abstract transition model and feasibility prediction model: $s'\_{i+1} = \mathcal{T}'(s'\_{i}, a'\_i)$ and $\overline{\textit{feas}}\_i = f(s'\_i, a'\_i)$, for all $i=k,\cdots,\ell-1$.
> - **Objective**: The training loss is defined as $\mathcal{L} = \sum\_{i=k+1}^{\ell} \| s'\_i - \phi(s\_i) \|\_2 + \sum_{i=k}^{\ell-1} \text{BCE}(\overline{\textit{feas}}\_i, \textit{feas}\_i)$, where $\text{BCE}$ is the binary cross-entropy loss, and $\textit{feas}\_i$ is the groundtruth feasibility label. $\textit{feas}\_i$ is true for all $i < \ell$, and is true for $i=\ell$ if the trajectory execution is successful (i.e., the last action is successful) or false otherwise.
> - **Model**: A single model is learned for all abstract actions because the abstract feasibility prediction model $f(s', a')$ and the abstract transition model $\mathcal{T}'(s', a')$ are both conditioned on the given abstract actions. The Point Cloud Transformer is trained as part of the state abstraction function $\phi$ with the loss described above.
> - **Data**: For BabyAI, we train all models on 100,000 trajectories. For Kitchen-Worlds, we train all models on around 10,000 trajectories. We provide details in Appendix C.
>
> **R1.5:** Details for planning (Section 4.3).
>
> **A:** We always choose the sequence of abstract actions with the highest feasibility scores. This design ensures that the agent will always pick an action at any state. To deal with situations where the action selected by the agent is not helpful for reaching the goal, we perform a close-loop replan after the agent finishes executing each action, therefore allowing the agent to recover.
>
> In this paper, we assume that the set of actions derived from language annotations of the training trajectories covers the space of actions required to complete the testing tasks. As discussed in Sec 4.1, in contrast to representing each action term as a textual sentence, the decomposed action and object concepts allow us to enumerate all possible actions an agent can take easily. In the experiments, we empirically validate whether our method can generalize to new combinations of action and object concepts in by evaluating our model on "Novel Concept Combinations".
>
> We note that it's non-trivial to generalize a manipulation policy to completely new actions (e.g., from pick-and-place to opening a microwave door). As discussed in a recent paper representative of the state of generalizable manipulation [A2], there are four levels of generalization: (1) different initial object placements, (2) different background and distractor objects, (3) new object-skill combinations, (4) new environment layouts. In this paper, our method shows strong generalization in all four levels.
>
> [A2] Bharadhwaj et al. Roboagent: Generalization and efficiency in robot manipulation via semantic augmentations and action chunking. arXiv:2309.01918, 2023.

---

> > ### Author Response · Authors · 2023-11-21
> > **Response Part 3**
> >
> > **R1.6:** Details for low-level policy (Section 4.4).
> >
> > **A:** Our low-level policy is trained with the advantage actor-critic (A2C) algorithm, following the same design as the baselines studied in the original BabyAI paper [A3]. The observation is a local 7x7 grid in a partial and egocentric view of the agent. The action and object concepts are encoded with learned word embeddings. The model uses FiLM layers [A4] to fuse the features from the state and the action concepts. Both the actor and the critic model use the same FiLM-style design. There are seven low-level actions: turn left, turn right, move forward, pick up an object, drop an object, toggle and done. A reward ranging from 0 to 1 is given to the agent only when it achieves the goal; the quicker the agent completes an episode, the closer to 1 the reward will be.
> >
> > [A3] Chevalier-Boisvert et al. BabyAI: A Platform to Study the Sample Efficiency of Grounded Language Learning. ICLR 2019.
> > [A4] Perez et al. FiLM: Visual Reasoning with a General Conditioning Layer. AAAI 2018.
> >
> > **R1.7:** Comparison with BabyAI expriments in original paper.
> >
> > **A:** Thank you for the suggestions. We have included a new result on training an end-to-end policy using imitation learning. Due to the time limit, we are only testing it on a single environment; we will include all variations in future revisions. The following table summarizes the result:
> >
> > |              | End-to-End RL | End-to-End IL | Ours (high + low) | High-Level RL | High-Level IL | Ours (high only) |
> > | ------------ | ------------- | ------------- | ------------------ | ------------- | ------------- | ---------------- |
> > | Key-Door     | 0             | 0             | 0.45               | 0.27          | 0.47          | 0.87             |
> >
> > You are indeed correct that we are using a similar neural network architecture as the methods studied in the original BabyAI paper. However, there are two key differences:
> >
> > - We focus on testing the generalization of models to, for example, novel concept combinations and longer high-level steps. By contrast, in the original BabyAI paper, methods are mostly trained and tested on similar environments. For example, both the End-to-End IL and the High-Level IL are capable of achieving a 0.99 success rate in the training environments (containing at most 3 high-level steps). However, there is a significant performance drop on generalization tests.
> > - The generalization-to-longer-step environments (e.g., Key-Door) require a significantly longer number of high-level steps than the environments studied in the original paper. For example, in the original paper, in the "boss" level, the agent only need to unlock at most one door to reach the target object (therefore, the number of high-level steps is 3). By contrast, in Key-Door, we are training on instances with at most 3 high-level actions, but testing them on scenarios with at least 5 steps (unlocking two doors). It makes sense for the baseline model to fail in all the tasks.
> >
> > Regarding your particular questions on details:
> > - **Sample efficiency**: We use 100,000 episodes of demonstration for imitation learning training, which is the same for all methods (baselines and ours). This is smaller than the number of episodes that the original BabyAI paper uses to train the models (1,000,000 episodes).
> > - **Performance in training environments**: Some of the baselines are capable of achieving good (i.e., >90%) success rates in training environments. However, they fail to generalize to more complex tasks.
> > - **Action spaces**: The high-level plans are compositions of one action name (e.g., pick-up), and two object concepts (e.g., red, key). In the **Key-Door** and **Two-Corridors** setting, the object type are constrained to key or door, and in the **All-Objects** setting, the object can be any type. Hence, there are 24 unique abstract actions in the **Key-Door** and Two-Corridors settings and 48 unique abstract actions in the **All-Objects** setting. We now discuss the involved action and object concepts for both environments in detail in Appendix C.
> >
> > We have also included new visualizations of the high-level plans. Please kindly refer to our updated paper (Appendix D.1) for details.

---

> > > ### Author Response · Authors · 2023-11-21
> > > **Response Part 4**
> > >
> > > **R1.8:** Details for Kitchen-Worlds experiments.
> > >
> > > **A:** In Kitchen-Worlds, there are in total 360 unique abstract actions (including combinations of actions, colors, object categories, and locations). We use around 10,000 trajectories to train the model, with details presented in Appendix C. In the **All-Objects** task setting, each plan consists of two abstract actions. In the **Sink** task setting, each plan consists of up to four abstract actions. An example of the predicted plan for the **Sink** task setting is presented in Figure 7. The goal is to place the blue mug into the sink. However, since the pan in the sink takes up too much space in the sink, a successful plan would include four abstraction actions: (pick-up, blue, pan, sink) (place, blue, pan, left counter) (pick-up, blue, mug, right counter) (place, blue, mug, sink). The low-level policies output either an object to pick or an object to drop the current holding object on. Picking is implemented as choosing a stable grasp on the target object from all stable grasps in the scene. We did not train our own grasp samplers; instead, we implemented the system using a task-agnostic grasp sampler based on ContactGraspNet [A5]. Placement is implemented as generating a placement pose for the object based on its 3D geometry within the region of the target location.
> > >
> > > [A5] Sundermeyer, et al. Contact-graspnet: Efficient 6-dof grasp generation in cluttered scenes. ICRA, 2021.

---

> > > > ### Comment · Reviewer_Q2SJ · 2023-11-22
> > > >
> > > > Dear Authors,
> > > >
> > > > I need more time to revisit the revised version of the paper and the details in the response.
> > > > Thanks very much for your response and the updates to the paper.

---

### Author Response · Authors · 2023-11-21
**General Response**

We thank all the reviewers for their insightful comments. We are glad everyone appreciates PARL’s novelty. All the feedback has helped us significantly improve the paper.

Updates for the rebuttal:

- We have made significant changes to the paper to help clarify our problem formulation and method. The changes in the revised paper are highlighted in blue.
- We have added five new sections in the appendix to provide more detailed discussions about using LLMs, network architectures for both the 2D and 3D environments, baseline implementations, data collection and evaluation procedures, additional results and visualizations, and our assumptions. We hope these details improve the credibility of the empirical results and reproducibility of our approach.
- We have performed additional experiments with the imitation learning (IL) baselines from the original BabyAI paper on the challenging Key-Door setting. We are happy to report that our method obtains a 40% improvement in high-level planning and a 45% improvement in combined high-level and low-level executions.

---

### Meta-Review · Area_Chair_kdA4 · 2023-12-20

**Metareview:**

The paper presents a method for solving planning problems by combining high-level actions generated by language models, and low-level policies. Reviewers appreciated its interesting combination of existing components, generality of the method, and potential impact in the space of language-based agents. However, concerns were raised about the insufficient experiment details, and a more thorough comparison to existing methods. Nonetheless, I recommend acceptance as I believe the authors adequately addressed the issues raised by reviewers.

**Justification For Why Not Higher Score:**

Relative lack of novelty and thorough experiment details.

**Justification For Why Not Lower Score:**

General and interesting combination of existing components.

---

### Decision · Program_Chairs · 2024-01-16

Accept (poster)